# Understanding the Interactions between *Staphylococcus aureus* and the Raw-Meat-Processing Environment Isolate *Klebsiella oxytoca* in Dual-Species Biofilms via Discovering an Altered Metabolic Profile

**DOI:** 10.3390/microorganisms9040672

**Published:** 2021-03-24

**Authors:** Xiaoxue Chen, Yunan Hu, Simin Tian, Beizhong Han

**Affiliations:** Beijing Laboratory for Food Quality and Safety, College of Food Science and Nutritional Engineering, China Agricultural University, Beijing 100083, China; chen.xx@cau.edu.cn (X.C.); 2017306100305@cau.edu.cn (Y.H.); 15662009961@163.com (S.T.)

**Keywords:** dual-species biofilms, *Klebsiella oxytoca*, *Staphylococcus aureus*, interactions, metabolomics

## Abstract

In a raw-meat-processing environment, members of the Enterobacteriaceae family can coexist with *Staphylococcus aureus* to form dual-species biofilms, leading to a higher risk of food contamination. However, very little is known about the effect of inter-species interactions on dual-species biofilm formation. The aim of this study was to investigate the interactions between *S. aureus* and raw-meat-processing environment isolates of *Klebsiella oxytoca* in dual-species biofilms, by employing an untargeted metabolomics tool. Crystal violet staining assay showed that the biomass of the dual-species biofilm significantly increased and reached its maximum after incubation for 21 h, compared with that of single species grown alone. The number of *K. oxytoca* in the dual-species biofilm was significantly higher than that of *S. aureus*. Field emission scanning electron microscopy (FESEM) revealed that both species were evenly distributed, and were tightly wrapped by extracellular polymeric substances in the dual-species biofilms. Ultra-high-pressure liquid chromatography equipped with a quadrupole-time-of-flight mass spectrometer (UHPLC-Q-TOF MS) analysis exhibited a total of 8184 positive ions, and 6294 negative ions were obtained from all test samples. Multivariate data analysis further described altered metabolic profiling between mono- and dual-species biofilms. Further, 18 and 21 different metabolites in the dual-species biofilm were screened as biomarkers by comparing the mono-species biofilms of *S. aureus* and *K. oxytoca*, respectively. The Kyoto Encyclopedia of Genes and Genomes (KEGG) pathways that were exclusively upregulated in the dual-species biofilm included ABC transporters, amino acid metabolism, and the two-component signal transduction system. Our results contribute to a better understanding of the interactive behavior of inter-species biofilm communities, by discovering altered metabolic profiling.

## 1. Introduction

Multiple microbes are important contributors to food spoilage during storage, processing, and distribution. These microbes coexist as communities, compete for resources and nutrients, and are often present as biofilms in the food-processing environment [1,2]. Biofilms are sessile microbial communities that rest in a hydrated matrix of self-produced extracellular polymeric substances, comprised of polysaccharides, proteins, and nucleic acids [3]. Biofilm formation is generally a dynamic process, and different mechanisms are involved in attachment and biofilm maturation. *Staphylococcus aureus*, a strain of Gram-positive bacteria, is an important commensal opportunistic and life-threatening food-borne pathogen because of its ability to produce staphylococcal enterotoxins in food [4]. The growth of *S. aureus* cells occurs in two states, the planktonic and biofilm states, with different specifications. Mature *S. aureus* biofilms are more resistant to environmental stress than their free-living counterparts [5].

Meat waste and losses are associated with spoilage microorganisms, which are usually sourced from biofilm cross-contamination [6]. Bacterial biofilms that are formed on equipment surfaces are potential sources of cross-contamination, and can be responsible for the spread of bacteria involved in food spoilage. These bacteria include some Enterobacteriaceae family members such as *Klebsiella* spp., which can cause a variety of nosocomial and foodborne infections [7,8]. *Klebsiella oxytoca* (*K. oxytoca*) is a coliform that is commonly found in the environment and in animal feces, vegetables, and pathological processes [9,10]. Gundogan, Citak and Yalcin [8] reported that, of the 45 *Klebsiella* species isolated from 60 calf and chicken meat samples collected from various supermarkets in Ankara, Turkey, 53% were identified as *K. oxytoca*, while all isolates were resistant to two or more antimicrobial agents [8]. They also suggested that meat and its products represent potential hazardous sources of multidrug-resistant and virulent *Klebsiella* species. *K. oxytoca* isolated from spoiled chicken carcasses showed high adhesion capacity on polystyrene surfaces at 20 °C, and on stainless steel surfaces under short-term attachment scenarios [6].

Most environmental biofilms contain a variety of microbes that can establish cooperative and competitive interactions, possibly resulting in an increase or decrease in antimicrobial resistance [11]. Numerous studies have been conducted on various aspects of biofilm formation by *S. aureus*, with most investigations focusing on monocultures. However, very few studies have focused on the real-world situations encountered by microbes, such as those found in food processing environments. In these processing environments, *K. oxytoca* can be present on equipment and other surfaces within these facilities, along with a variety of different bacteria. Study on bacterial interactions between *S. aureus* and *K. oxytoca* may reveal a vital feature of interspecies relationships in biofilms, such as their influence on biofilm metabolism and antimicrobial resistance. Hence, in our current study, crystal violet staining was employed to characterize the biofilm formation of mono- and dual-species of *S. aureus* and *K. oxytoca* biofilms. In addition, we used an untargeted metabolomics tool to investigate the metabolic profiles and differential metabolites in the interactions between mono- or dual-species biofilms (DSBs) formed by *S. aureus* and *K. oxytoca*. The aim of this study was to explore their interactions in biofilms, and lay a theoretical foundation for the development of new methods to control DSBs in the food industry.

## 2. Materials and Methods

### 2.1. Bacterial Strains and Media

We used the strain *K. oxytoca* 88, a strong spoilage organism previously isolated from food production environments, and the standard strain *S. aureus* ATCC 6538, to test DSBs in this study. The nucleotide sequence of *K. oxytoca* 88 was placed in GenBank database under accession number SUB9024481 KO88 MW559226. The cultures were stored in Nutrient Broth (NB) with 30% glycerol (*v/v*) at −80 °C in our laboratory. Before utilization, the strains were incubated in NB solution at 37 °C with shaking at 150 rpm for 12 h.

### 2.2. Development of Mono- and Dual-Species Biofilms and Quantification

For biofilm formation, the bacterial suspension containing NB (approximately 10^7–8^ CFU/mL) was individually added into each well of a 96-well polystyrene plate (Corning, NY, USA). The mono-species biofilms were established with 200 μL of suspension in each well, and the corresponding strain and DSBs were established with 100 μL of each bacterial suspension. After that, the biofilms were incubated at 37 °C without shaking to allow biofilm formation for 6, 9, 12, 15, 18, 21, and 24 h. After incubation, the supernatants were removed following washing with sterile PBS to each well, to remove the loosely attached cells. The air-dried samples were fixed with methanol and then stained with 200 μL of 1% crystal violet solution for 10 min. Finally, the stained samples were dissolved in 200 μL of 33% acetic acid. The biomass of the biofilms was quantified by reading the microplates at 595 nm using the Multiskan Spectrum (Thermo Fisher, Waltham, MA, USA). The viability of the total bacteria in the biofilm was characterized by using a cell counting kit-8 (CCK-8; Solarbio, Beijing, China) and measuring the absorbance at 450 nm with a microplate reader (Thermo Fisher, Waltham, MA, USA). In addition, to enumerate the biofilm cells, the plate was incubated for 6, 12, and 21 h. The wells were then carefully washed three times with PBS to release loosely attached cells. Subsequently, the biofilm cells were harvested in a sterile tube containing 6 mL 0.85% saline solution, and vortexed for 30 s. Baird-Parker Agar (BPA) and Violet Red Bile Agar (VRBA) were used as the selective media to enumerate *S. aureus* and *K. oxytoca* colonies in the DSB, respectively. All agar plates were incubated at 37 °C for 24 h before manual enumeration of the colonies.

### 2.3. Field Emission Scanning Electron Microscope (FESEM) Analysis

For FESEM analysis, the mature mono- and dual-species biofilms were established on glass coverslip in 6-well polystyrene plates (Corning, NY, USA) covered, as described above. Following biofilm formation, the harvested mono- and dual-species biofilms were washed three times with PBS and then fixed with 2.5% glutaraldehyde for 6 h at 4 °C. Subsequently, the samples were gradually dehydrated in graded ethanol (25, 50, 75, 90, 95, and 100%) for 10 min and then air-dried, gold-coated, and examined under a FESEM (Quanta FEG 650, FEI, Hillsboro, OR, USA).

### 2.4. Biofilm Collection and Metabolite Extraction

The mature mono- and dual-species biofilms were established on 6-well polystyrene plates (Corning, NY, USA) as described above. After 21 h of incubation, the planktonic cells were removed with sterilized PBS. The biofilms were incubated in sterile saline solution for another 3 h at room temperature. The mixtures were subsequently sonicated and then centrifuged for 3 min at 4 °C to harvest the biofilm samples [12]. The supernatants were mixed with 1 mL of pre-cooled methanol/acetonitrile/water solution (2:2:1, *v*/*v*) successively for 1 min of vortexing, and then the mixtures were sonicated at low temperature for 30 min. Subsequently, the samples were allowed to stand at –20 °C for 60 min, followed by centrifugation at 14,000× *g* for 20 min at 4 °C. The supernatant was collected and vacuum freeze-dried for sample analysis. Finally, dried samples were dissolved in 100 µL acetonitrile/water (1:1, *v*/*v*), followed by vortexing and ultrasonic treatments. Next, each sample was centrifuged at 14,000× *g* at 4 °C for 15 min, and then each sample was transferred to a sample bottle with a lining tube for subsequent chemical analysis.

### 2.5. HILIC UHPLC-Q-TOF MS Analysis

The untargeted metabolite profiling of the mono- and dual-species biofilms was analyzed by means of ultra-high-pressure liquid chromatography (UHPLC; Agilent Technologies, Santa Clara, CA, USA) with a quadrupole-time-of-flight mass spectrometer (QTOF-MS) system (AB Sciex TripleTOF 5600, Framingham, MA, USA). For hydrophilic interaction liquid chromatography (HILIC) separation, metabolites were separated using a Waters ACQUITY UPLC BEH Amide column (1.7 µm, 2.1 × 100 mm) at 25 °C. The flow rate was 0.3 mL/min and the mobile phases were composed of phase A (25 mM ammonium acetate and 25 mM ammonia in water) and phase B (acetonitrile). The gradient was 95% B for 1 min and was linearly reduced to 65% at 14 min, followed by reduction to 40% at 18–23 min. The gradient was then increased to 95% and maintained. Both positive ion and negative ion modes were used for sample detection via electrospray ionization (ESI). The samples separated by UHPLC were analyzed using a Triple TOF 5600 mass spectrometer. The ESI source conditions were set as follows: Ion Source Gas1: 60; Ion Source Gas2: 60; Curtain gas: 30; source temperature: 600 °C; and Ion Sapary Voltage Floating: as ±5500 V (both positive and negative modes). The TOF MS scan *m*/*z* range was 60–1000 Da, the product ion scan *m*/*z* range was set as 25–1000 Da, the TOF MS scan accumulation time was set as 0.20 s/spectra, and the product ion scan accumulation time was set at 0.05 s/spectra. The secondary mass spectrum was obtained by information-dependent acquisition, and the high sensitivity mode was adopted. The declustering potential was set at ±60 V (both positive and negative modes) and the collision energy was 35 ± 15 eV. The IDA settings were set as isotopes within 4 Da, and with six candidate ions to monitor per cycle.

Quality control (QC) samples were prepared by pooling aliquots of all samples that were representative of the mono- and dual-species biofilms under analysis for data normalization.

### 2.6. Multivariate Data Processing and Data Analysis

The raw dataset obtained by HILIC UHPLC-Q-TOF MS was converted into common data format (.mzXML) files using Proteo Wizard software (Palo Alto, CA, USA). The XCMS program was then used for peak alignment, retention time correction, and peak area extraction. The identification of the metabolite structure adopted the accurate mass matching (<25 ppm) and secondary spectrum matching methods by searching the self-built database of the laboratory.

For the dataset extracted by XCMS, ion peaks with missing values >50% within the group were deleted. Multi-dimensional statistical analyses, including unsupervised principal component analysis (PCA), Partial Least Squares Discrimination Analysis (PLS-DA), and Orthogonal Bias Discriminant Analysis of Least Squares (OPLS-DA), were performed by means of SIMCA-P 14.1 (Umetrics, Umea, Sweden). The model parameters *R*^2^*Y* and *Q*^2^*Y* were inspected to check the goodness of the prediction model. In addition, variable importance in projection (VIP) analysis was conducted on the processed data using the standard algorithms in the PLS-DA toolbox. Metabolites with a VIP value >1 were regarded as the most influential factors in the extracted PLS-DA models. A volcano plot combining fold change analysis (FC > 2) and the *t*-test (*p* < 0.05) was performed, considering the UHPLC-Q-TOF MS data to provide the up/down accumulation trends of the discriminant markers gained by multivariate statistics. Finally, the related metabolic pathways were analyzed based on the Kyoto Encyclopedia of Genes and Genomes (KEGG) database. The basic significance statistics were carried out by one-way analysis of variance (ANOVA) with SPSS software (Version 19.0; Chicago, IL, USA). Differences were considered statistically significant at *p* < 0.05.

## 3. Results

### 3.1. Characterization of Mono- and Dual-Species Biofilms

The growth models of mono- and dual-species biofilms are shown in Figure 1A. The mono- and dual-species biofilms formed by *S. aureus* and *K. oxytoca* exhibited different growth dynamics. The biomass of the DSB formed by *S. aureus* and *K. oxytoca* was significantly higher than those of the two monospecies biofilms formed after the 12 h incubation period (*p* < 0.05). The biomass of the DSB reached its highest at 21 h. At this time, the biomasses of the monospecies biofilms formed by *S. aureus* and *K. oxytoca* were 53.38% and 62.74% of the DSB, respectively. Figure 1B shows that the bacterial viability in the DBSs increased significantly from 9 h to 12 h, reaching their highest viability at 12 h and remaining stable. A significant decrease (*p* < 0.05) in bacterial viability was observed in the DSBs at 24 h. We enumerated the colony forming unit (CFU) numbers of both *S. aureus* and *K. oxytoca* strains in the DSBs. As shown in Figure 1C, the number of *K. oxytoca* and *S. aureus* increased during the incubation time of the DSBs, and the number of *K. oxytoca* was significantly higher than that of *S. aureus* (*p* < 0.05). Thus, the DSBs matured at 21 h, while the numbers of *K. oxytoca* and *S. aureus* in the biofilm were 7.9 and 6.8 log CFU/mL, respectively.

FESEM analysis was employed to further reveal the symbiotic state of *S. aureus* and *K. oxytoca* in DSBs. Each strain produced an apparent biofilm and an extracellular matrix in NB individually (Figure 2A,B). The FESEM images of the mature DSBs at 21 h showed the capability of *S. aureus* and *K. oxytoca* to adhere and form an even mixed biofilm on the glass coverslip (Figure 2C,D). We also found that after co-cultivation of *S. aureus* and *K. oxytoca*, the biofilm exhibited more extracellular matrix, suggesting that the synergistic growth of the two species promoted the formation of the DSB.

### 3.2. Metabolic Profiling by UHPLC-Q-TOF/MS

The peaks obtained from all samples and the QC samples were subjected to Pareto-scaling followed by PCA analysis. The results showed that the QC injections in different groups were clustered tightly in positive and negative ion modes (Figure 3A,B), indicating a satisfactory stability of the test system. Furthermore, according to the relative quantitative values of the metabolites, the Pearson correlation coefficients between all samples were calculated. A higher correlation coefficient (R^2^ was close to 1) indicated better stability of the entire detection process and higher data quality (Figure 3C,D).

OPLS-DA is a statistical method of supervised discriminant analysis. The parameters of the OPLS-DA model were obtained by 7-fold cross-validation to evaluate the fitness and prediction ability of our model. Generally, the *R*^2^*Y* and *Q*^2^ were ≥ 0.5, suggesting that the model was stable and reliable.

Thus, we used OPLS-DA analysis to further evaluate the difference between the metabolite profiles of different groups (Figure 4). The OPLS-DA score plots and loading plots for the positive and negative modes between the *S. aureus* biofilm (SAB) and DSB were established. An obvious separation trend between the two groups was observed in the OPLS-DA score plot (Figure 4A,A1). The model parameters were *R*^2^*X* = 56.7%, *R*^2^*Y* = 99.9%, and *Q*^2^ = 80.4% in the positive ion model and *R*^2^*X* = 80.4%, *R*^2^*Y* = 100%, and *Q*^2^ = 99.7% in the negative ion model. These results suggest that the model has good fitness and predictive ability. For the OPLS-DA score plots and loading plots in the positive and negative modes between the *K. oxytoca* biofilm (KOB) and DSB, we can also see clear separation trends in both the positive and negative ion modes. The model parameters were *R*^2^*X* = 49.8%, *R*^2^*Y* = 100%, and *Q*^2^ = 97.7% in the positive ion model and *R*^2^*X* = 84%, *R*^2^*Y* = 100%, and *Q*^2^ = 99.6% in the negative ion model. These results indicated that the biofilm metabolic profiles differed between the mono- and dual-species biofilms.

### 3.3. Screening of Potential Biomarkers

Univariate analysis methods including fold change analysis, *t*-tests, and volcano plots are widely employed to analyze the difference in metabolite profiles between two samples. Univariate analysis can visually display the significance of metabolite changes between two groups, which is conducive to further screening for potential marker metabolites. In this study, we used volcano plot analysis to reveal significantly altered metabolite features between mono- and dual-species biofilms formed by *S. aureus* and *K. oxytoca* in the positive and negative ion mode, by delineating a log transformation plot of the fold-change difference and the level of statistical significance of each metabolite.

As shown in Figure 5A,C, the volcano plots expressing the differential metabolites showed significant differences between the SAB and DSB groups. Using FC value screening >2 at a level of *p* < 0.05, a total of 536 differential metabolites were identified in the positive ion mode. In detail, the DSB group exhibited a total of 309 upregulated metabolites and 227 downregulated metabolites relative to the SAB group (Figure 5A). Similarly, a total of 598 differential metabolites were screened in the negative ion mode. Of these metabolites, a total of 434 were upregulated and 164 were downregulated in the DSB group compared with those in the SAB group (Figure 5C). The volcano plots of differential metabolites in biofilm cells between the KOB and DSB in the positive and negative modes are shown in Figure 5B,D. A total of 743 differentially expressed metabolites were identified in the positive ion mode. Of these metabolites, a total of 420 were upregulated and 323 were downregulated in the DSB group compared with the KOB group (Figure 5B). In the negative ion mode, a total of 624 differential metabolites, including 398 upregulated and 226 downregulated metabolites, were identified in the DSB group compared with the KOB group (Figure 5D).

The identifying information of each differentially expressed metabolite under the positive and negative ion modes of the DSB compared to SAB are provided in Table 1. As shown in Table 1, 18 different metabolites highly associated with biofilm formation that were significantly upregulated in the DSB were selected. Among the differential metabolites selected, 13 metabolites were upregulated for the positive (ESI+)/negative (ESI–) in the DSB compared to SAB, and five metabolites were downregulated for the ESI+/ESI–. In the ESI+ mode, differential metabolites of L-phenylalanine, L-glutamate, L-alanine, L-isoleucine, L-arginine, L-valine, raffinose, L-glutamine, betaine, and phosphorylcholine were obtained as the key biomarkers to distinguish the difference between DSBs and SABs. However, L-phenylalanine, citraconic acid, glycerol 3-phosphate, citrate, L-lactate, L-leucine, myo-inositol, and L-histidine were obtained as the key biomarkers in ESI– mode.

Table 2 shows the identifying information of each differentially expressed metabolite under the positive and negative ion modes of the DSB compared to KOB. There were 21 selected differential metabolites related to biofilm formation in the DSB compared with the KOB (Table 2). In the DSB, the levels of 14 differential metabolites were higher, while 7 metabolites were lower, than those in the KOB for ESI+/ESI. Among them, L-arginine, hydroxyproline, D-proline, raffinose, N-acetylputrescine, L-alanine, Glycine, L-aspartate, 4-Guanidinobutyric acid, sucrose, and L-glutamine were obtained in ESI+ mode. L-arginine, hydroxyproline, sarcosine, L-leucine, DL-lactate, succinate, sucrose, L-histidine, citrate, and L-valine were obtained in ESI– mode. These differential metabolites can be considered as key biomarkers to distinguish between the DSB and KOB.

### 3.4. KEGG Pathway Enrichment Analysis

To evaluate the most relevant pathways and the potential mechanisms involved in DSB formation, the KEGG database was used to carry out pathway enrichment analysis, and the significance of the enriched pathways was calculated by Fisher’s exact test. Details of the top 30 metabolic pathways that were differentially expressed in the DSB vs. SAB and DSB vs. KOB are given in Figure 6. The larger the Rich factor value, the higher the enrichment of differential metabolites in the pathway. The color of the dot represents the *p*-value of the hypergeometric test, suggesting that the test was reliable and statistically significant. The size of the dot represents the number of different metabolites in the corresponding pathway.

As shown in Figure 6A, the results suggested that these differential metabolites between the DSB and SAB are involved in key metabolic pathways, including ABC transporters, arginine and proline metabolism, valine, leucine and isoleucine biosynthesis, arginine biosynthesis, and alanine, aspartate and glutamate metabolism. Figure 6B reveals that multiple key biochemical pathways are related to these differential metabolites between the DSB and KOB, including ABC transporters, alanine, aspartate and glutamate metabolism, arginine and proline metabolism, arginine biosynthesis, galactose metabolism, histidine metabolism, two-component system, D-alanine metabolism, and glyoxylate and dicarboxylate metabolism.

## 4. Discussion

The Enterobacteriaceae family is regarded as a hygiene indicator in food processing, and members of this family are among the most challenging food-borne pathogens that contaminate vegetables, raw and processed meat products, and fish [7]. The pattern of biofilms represents the dominant life form of most microorganisms in the environment [13]. Numerous investigations into biofilms have been conducted using monospecies, but biofilms generally comprise a mixture of species. Interactions among species have a vital role in the formation, structure, and function of the biofilm. These interactions are roughly categorized as either cooperative or competitive, depending on the underlying type of social behavior and molecular mechanisms [14]. In meat processing environments, environmental colonizing by Enterobacteriaceae family members is most likely to form mixed-species biofilms with other pathogens, such as *S. aureus*, which has been linked to the causative agent of food poisoning [3,15].

*S*. *aureus* is a Gram-positive bacterium, while *K*. *oxytoca* is a Gram-negative bacterium, and they both can form biofilms that cause serious food safety problems in the food industry [3,8,10]. Specifically, *K. oxytoca* is one of the most frequently isolated strong biofilm-forming bacterial species from raw-meat processing environments [6]. Tang, Flint, Bennett, Brooks, and Morton (2009) investigated the biofilm growth of *K. oxytoca* strains K. B006 and TR002, retrieved from the dairy industry, on ultrafiltration membranes, both individually and together [16]. It was found that the dual strains produced a higher biofilm density on the new membranes than a single strain. However, the dynamics of the DSB population remain unclear, and the interaction of *S. aureus* with *K. oxytoca* in a DSB has never been studied. In this investigation, we found that *K. oxytoca* isolated from the raw meat processing environment exhibited a strong ability to promote *S. aureus* biofilm formation. Hence, the present study employed an untargeted metabolomics approach to investigate the effect of interspecies interactions on the ability of *S. aureus* and *K. oxytoca* to develop DSBs.

The specific spatial arrangement of the mono- and dual-species biofilms was visualized using FESEM after rinsing and fixation. We observed that *S. aureus* and *K. oxytoca* were evenly distributed in the DSB, and were tightly wrapped by extracellular polymeric substances. Based on these observations, we suggest that these two bacterial species interact with each other and have synergistic effects in forming DSBs. Burmolle et al. (2006) reported that the synergistic effect of microorganisms in multi-species biofilms can enhance the biomass of biofilms, the resistance of biofilms to antimicrobial agents, and bacterial invasion [17].

Metabolomics is capable of comprehensive qualitative and quantitative analysis of all low molecular weight metabolites present in and around the growing cells at a given time in the growth or production cycle [18]. This approach had received increased attention in recent years, and may lead to the discovery of unique metabolic pathways. Furthermore, metabolomics can help us to understand whether the interaction within the microbial community is a result of a synergistic or antagonistic relationship [19]. Untargeted metabolomics can maximize the chances of identifying compounds by simultaneously detecting as many metabolites as possible. Therefore, this method is more likely to unearth the biochemical pathways that have not been explored before under specific biological conditions [20]. We sought to understand the potential mechanism by which DSBs are formed, in addition to the synergistic effects between *S. aureus* and *K. oxytoca*. In this study, we employed UHPLC-Q-TOF MS to perform untargeted metabolomics analysis to discover altered metabolic profiles in mono- and dual-species biofilms [21]. We employed XCMS software to extract the ion peaks of metabolites, and found a total of 8184 positive ions and 6294 negative ions for all test samples. We described the changes in metabolic profiling between mono- and dual-species biofilm administration using an UPLC-QTOF-MS tool. Furthermore, we used screening conditions of VIP > 1 and a *p*-value < 0.05 in the positive and negative-ion modes to identify differentially expressed metabolites of the DSB, compared to those of the monospecies biofilms.

We identified and screened a total of 18 metabolites (nine L-amino acids, raffinose, betaine, phosphorylcholine, citraconic acid, glycerol 3-phosphate, citrate, DL-lactate, myo-Inositol, and L-histidine), mainly involved with ABC transporters, aminoacyl-tRNA biosynthesis, arginine and proline metabolism, mineral absorption, cyanoamino acid metabolism, valine, leucine and isoleucine biosynthesis, tyrosine metabolism, alanine, aspartate and glutamate metabolism, the two-component system, and the citrate cycle (TCA cycle) in the SAB and DSB in the positive and negative ion modes. In addition, a total of 21 metabolites (7 L-amino acids, hydroxyproline, D-proline, raffinose, N-Acetylputrescine, glycine, 4-Guanidinobutyric acid, sucrose, hydroxyproline, sarcosine, DL-lactate, succinate, sucrose, and citrate), mainly associated with ABC transporters, aminoacyl-tRNA biosynthesis, alanine, aspartate and glutamate metabolism, mineral absorption, arginine and proline metabolism, arginine biosynthesis, histidine metabolism, galactose metabolism, the two-component system, and D-alanine metabolism, were identified and screened in the KSB and DSB in the positive and negative ion modes. Similarly, Sadiq, Flint, Sakandar, and He (2019) revealed the altered transcription profile during the motile-to-sessile switch in high and low biofilm-forming *Bacillus licheniformis* isolated from Chinese milk powders [22]. The KEGG pathways, including valine, leucine and isoleucine biosynthesis and degradation, ABC transporters, and the metabolism of amino sugar and nucleotide sugar, were exclusively upregulated in the high biofilm former.

Amino acids (AAs) serve as intermediates in metabolism and exist predominantly in the L-isomeric form in all living cells, and play critical roles as building blocks for proteins [23]. In general, D-AAs, mainly produced by the amino acid racemases of microbes, are considered relevant constituents in some bacterial structures. Some D-AAs are associated with microbial growth fitness and processes such as biofilm formation, spore germination, and signaling [24]. L-AAs are the dominant substrates for ribosome-based protein synthesis [25]. Barrientos-Moreno, Molina-Henares, Ramos-Gonzalez, and Espinosa-Urgel found that both exogenous and endogenous L-arginine affect biofilm formation of *Pseudomonas putida* through changes in cyclic diguanylate content, and altered expression of structural elements of the biofilm extracellular matrix [26]. Velmourougane and Prasanna revealed that since L-AAs (mainly L-Glu, L-Gln, L-His, L-Ser, L-Thr and L-Trp) affect the growth, aggregation, and carbohydrate synthesis of microorganisms, they are essential for the formation of biofilms [23]. In this study, compared with the monospecies biofilms, numerous amino acids were detected in the DSBs. It can be inferred that the amino acid metabolism in a DSB is more active, which potentially contribute to the biofilm formation and maturation.

Polysaccharide intercellular adhesin (PIA) is distinctly related to the staphylococcal cell surface and mediates cell-to-cell adhesion, which is critical for mature biofilm formation [27]. Vuong et al. found that shunting carbon from the TCA cycle to the production of PIA is one of the effects of TCA cycle stress [27]. In the study of methicillin-sensitive *S. aureus*, it was found that the TCA cycle negatively regulates the production of extracellular polysaccharides (the main component in the biofilm matrix). When the TCA cycle is inactivated, it mainly affects the protein components in the biofilm matrix [28]. De Backer et al. found an important role of the TCA cycle in mediating biofilm formation, which affects the composition of the methicillin-sensitive *S. aureus* USA300 biofilm matrix [28]. Moreover, Shanks et al. reported that citrate promotes biofilm formation by stimulating both cell-to-surface and cell-to-cell interactions [29].

Differential metabolites, including citrate, succinate, L-aspartate and L-glutamine, are also related to the two-component signal transduction system (TCS). TCS is one of the dominating pathways by which bacteria adapt to the external environment, and it plays an important role in regulating the formation of bacterial biofilms, the expression of virulence genes, the synthesis of the cell wall, and bacterial activity [30]. Generally, multiple mechanisms exist in the TCS, such as cross-regulation, integration, and coordination of various input stimuli to control biofilm formation [31]. Sucrose, a substrate for the synthesis of extracellular polysaccharides but not oligosaccharides, is beneficial for biofilm formation. Nagasawa, Sato, and Senpuku reported that the cooperative effects of raffinose and sucrose influence *Streptococcus mutans* biofilm formation [32]. This effect mainly contributes to the production of extracellular DNA and fructan. Interestingly, we found that the sucrose content in the DSB was significantly higher than that in the KOB, but the raffinose content was lower in the DSB. Nuraini, Pradopo, and Pronorahardjo found that the adherence of sucrose-induced *S. mutans* biofilm is higher than that of a xylitol-induced *S. mutans* biofilm [33].

## 5. Conclusions

In conclusion, we revealed that the synergistic interaction between *S. aureus* and *K. oxytoca* significantly enhanced DSB formation. UPLC-ESI-Q-TOF-MS-based metabolomics profiling and multivariate statistical analysis indicated that 18 and 21 different metabolites in the DSB were biomarkers when compared with the monospecies SAB and KOB, respectively. These metabolites were mainly associated with ABC transporters, amino acid metabolism, and the TCS. Further research should elucidate the effects of different disinfectants on DSBs, and suggest appropriate control strategies for the contamination problem of dual-species biofilms in the food industry.

## Figures and Tables

**Figure 1 microorganisms-09-00672-f001:**
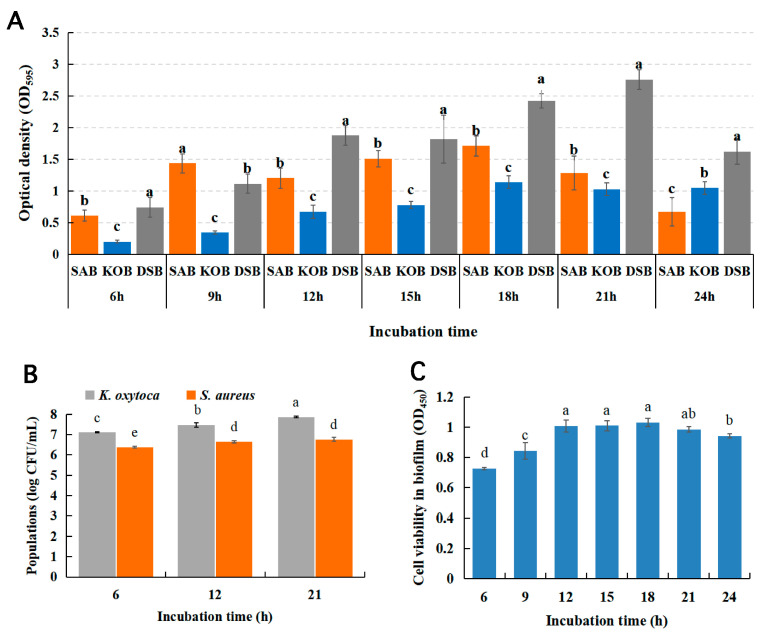
Biofilm growth model by crystal violet quantification of the *S. aureus* biofilm (SAB), *K. oxytoca* biofilm (KOB), and dual-species biofilms (DSBs) (**A**); Changes in bacterial viability in DSBs at different incubation times (**B**); The bacteria counts (log CFU/mL) of *S. aureus* and *K. oxytoca* in DSBs at 37 °C after 6, 12, and 21 h incubation (**C**). Different uppercase letters (a–e) show significant differences according to Duncan’s multiple-range test (*p* < 0.05). All experiments were repeated on three independent occasions, using eight wells per group.

**Figure 2 microorganisms-09-00672-f002:**
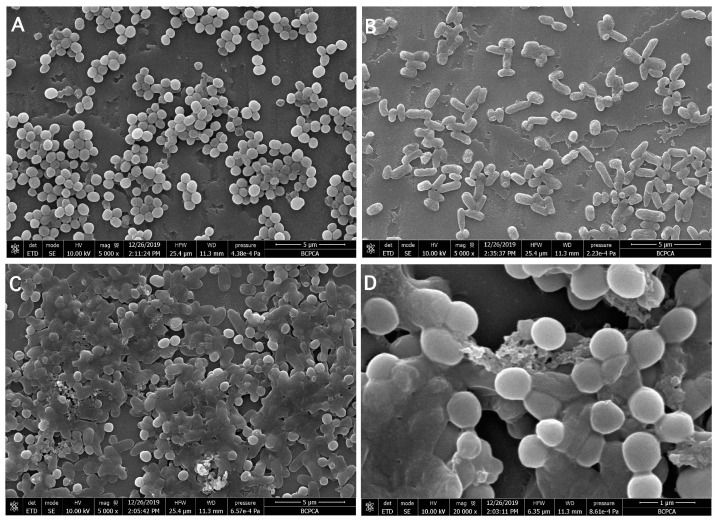
FESEM images showing the biofilm micrographs of mono- and dual-species biofilms formed by *S. aureus* and *K. oxytoca* for 21 h at 37 °C. (**A**) SAB, (**B**) KOB, (**C**,**D**) DSBs.

**Figure 3 microorganisms-09-00672-f003:**
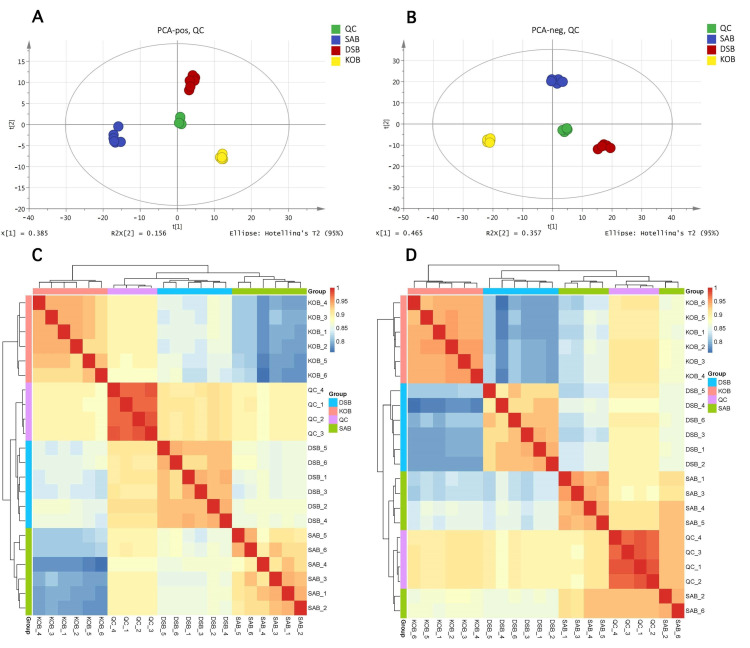
Principal components analysis (PCA) score plots of the SAB, KOB, and DSB in positive ion mode (**A**) and negative ion mode (**B**). The heatmap of the Pearson’s correlation analysis among all samples in positive ion mode (**C**) and negative ion mode (**D**). Each point represents an individual sample.

**Figure 4 microorganisms-09-00672-f004:**
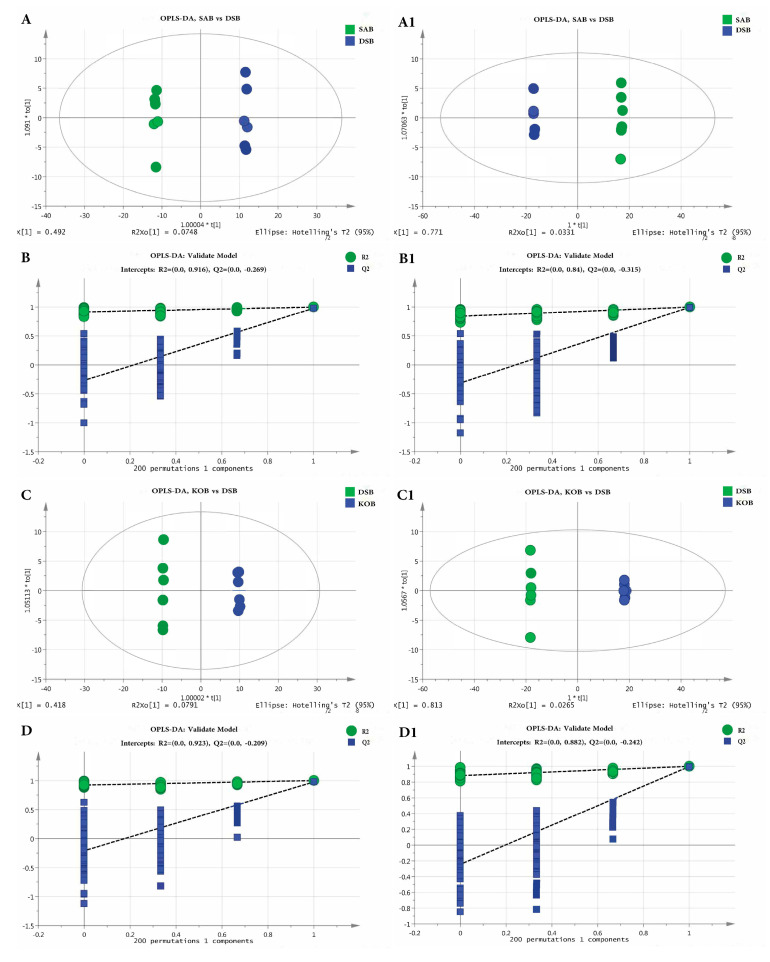
OPLS-DA analysis (**A**,**A1**) and OPLS-DA permutation test (**B**,**B1**) in positive (**A**,**B**) and negative (**A1**,**B1**) mode of the SAB vs. DSB, respectively. OPLS-DA analysis (**C**,**C1**) and OPLS-DA permutation test (**D**,**D1**) in positive (**C**,**D**) and negative (**C1**,**D1**) mode of the KOB vs. DSB, respectively.

**Figure 5 microorganisms-09-00672-f005:**
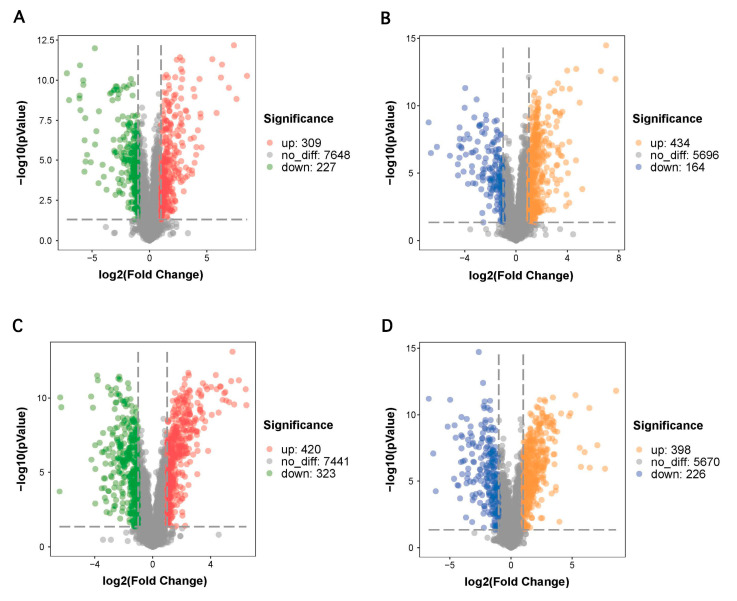
Volcano plots representing the significant differential metabolites in biofilms between the SAB and DSB in the positive (**A**) and negative (**B**) mode. The volcano plots of differentially expressed metabolites in biofilms between the KOB and DSB in the positive (**C**) and negative (**D**) mode.

**Figure 6 microorganisms-09-00672-f006:**
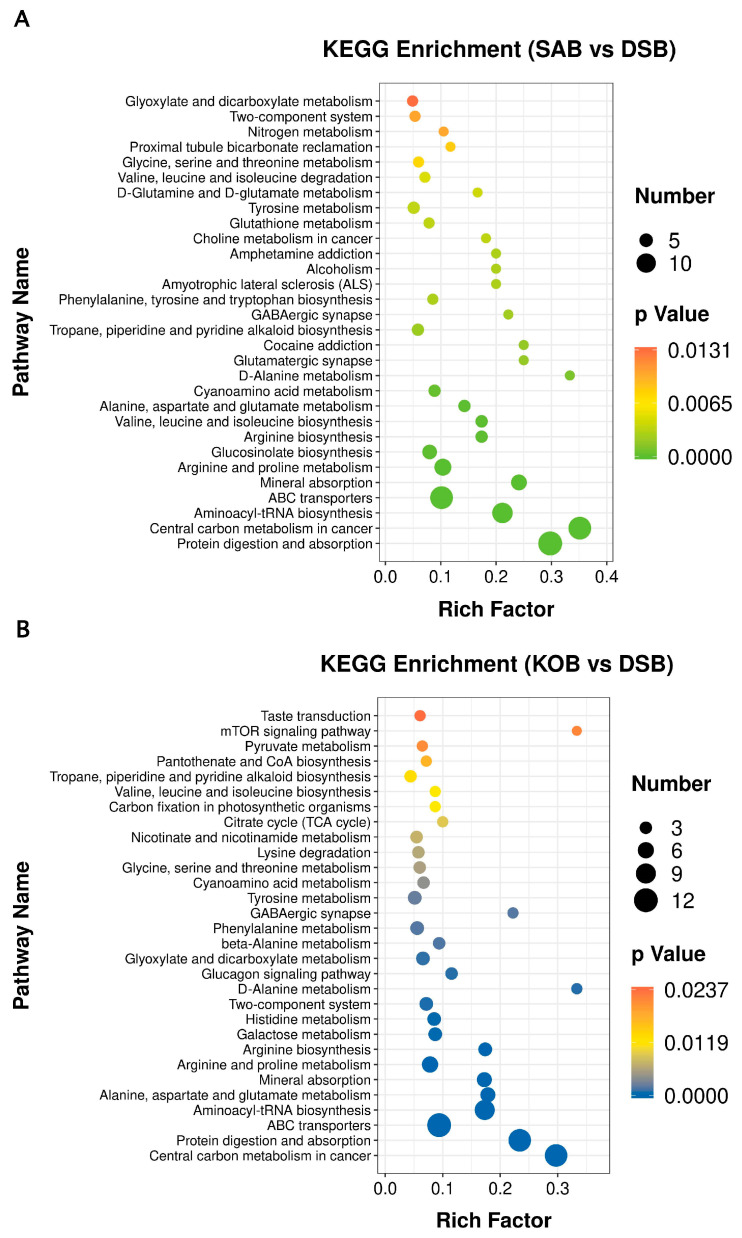
Top 30 KEGG enrichment pathways for differentially expressed metabolites in biofilms between the SAB and DSB (**A**) and in biofilms between the KOB and DSB (**B**).

**Table 1 microorganisms-09-00672-t001:** Some selected differential metabolites between the SAB and DSB.

Mode	No	Adduct	Rt(s)	*m*/*z*	Metabolite	VIP	Fold Change	*p*-Value
ESI (+)	1	(M + H) +	475.2005	166.0848377	L-phenylalanine	17.7719	7.253863106	0.000
	2	(M + H)+	770.426	148.0585192	L-glutamate	2.62697	6.590693393	0.000
	3	(M + H)+	651.8765	90.05302762	L-alanine	1.73871	3.383882682	0.011
	4	(M + H)+	510.724	132.1001379	L-isoleucine	6.32109	3.021382002	0.000
	5	(M + H)+	956.207	175.1174382	L-arginine	1.06786	1.463831102	0.036
	6	(M + H)+	565.397	118.084813	L-valine	3.23266	1.41510846	0.003
	7	(M + NH_4_)+	861.9085	522.2015198	Raffinose	1.47743	0.875635638	0.017
	8	(M + H-H_2_O)+	226.93	129.0638672	L-glutamine	1.84291	0.735582718	0.001
	9	(M + H)+	504.573	118.0849979	Betaine	7.42027	0.693008883	0.019
	10	(M + H)+	906.151	184.0714508	Phosphorylcholine	1.51653	0.214757887	0.000
ESI (–)	1	(M-H)-	470.852	164.0712635	L-phenylalanine	12.9064	11.88920095	0.000
	2	(M-H)-	745.958	129.0184177	Citraconic acid	1.24581	0.167383867	0.000
	3	(M-H_2_O-H)-	510.2385	152.9949551	Glycerol 3-phosphate	1.22255	3.659826379	0.000
	4	(M-H)-	900.126	191.0192075	Citrate	1.30013	4.143342579	0.000
	5	(M-H)-	428.5495	89.02377085	DL-lactate	2.68168	15.2069271	0.000
	6	(M-H)-	505.751	130.0866943	L-leucine	6.6272	1.700068517	0.002
	7	(M-H)-	744.368	179.0556474	Myo-inositol	1.14716	1.141321917	0.003
	8	(M-H)-	830.001	154.0615257	L-histidine	1.82022	3.032862317	0.008

**Table 2 microorganisms-09-00672-t002:** Some selected differential metabolites between the KOB and DSB.

Mode	No	Adduct	Rt(s)	*m*/*z*	Metabolite	VIP	Fold Change	*p*-Value
ESI (+)	1	(M + H)+	956.207	175.1174382	L-arginine	20.8719	86.28151666	0.000
	2	(M + H)+	651.4535	132.0640863	Hydroxyproline	7.84314	0.136447311	0.000
	3	(M + H)+	584.999	116.0687626	D-proline	9.17181	8.063013806	0.000
	4	(M+NH_4_)+	861.9085	522.2015198	Raffinose	4.22294	0.358515007	0.000
	5	(M + H)+	580.177	131.1162654	N-acetylputrescine	16.9024	0.167206532	0.000
	6	(M + H)+	651.8765	90.05302762	L-alanine	3.53333	7.029141017	0.000
	7	(M + H)+	684.3185	76.03758772	Glycine	2.0612	8.323423876	0.000
	8	(M + H)+	783.3	134.0430189	L-aspartate	1.29637	3.978271643	0.000
	9	(M + H)+	669.03	146.0908832	4-Guanidinobutyric acid	1.05007	0.786705284	0.010
	10	(M + NH_4_)+	738.9845	360.1485731	Sucrose	1.56376	1.879752696	0.010
	11	(M + H-H_2_O)+	226.93	129.0638672	L-glutamine	1.13001	1.155452429	0.034
ESI (–)	1	(M-H)-	982.1585	173.1038206	L-arginine	7.29498	84.61874522	0.000
	2	(M-H)-	645.4265	130.0499991	Hydroxyproline	1.44552	0.235656929	0.000
	3	(M-H)-	646.5645	88.03968055	Sarcosine	2.23067	11.0313482	0.000
	4	(M-H)-	483.812	130.086757	L-leucine	4.44469	0.092413693	0.000
	5	(M-H)-	428.5495	89.02377085	DL-lactate	5.46247	64.61340268	0.000
	6	(M-H)-	736.06	117.0184715	Succinate	5.20026	52.41791755	0.000
	7	(M-H)-	730.801	341.1082907	Sucrose	2.42076	7.285447139	0.000
	8	(M-H)-	830.001	154.0615257	L-histidine	2.048	3.51164159	0.000
	9	(M-H)-	959.703	191.0190107	Citrate	1.14805	15.54534738	0.011
	10	(M-H)-	555.6405	116.0709687	L-valine	1.63014	0.868488008	0.031

## Data Availability

The nucleotide sequence of *K. oxytoca* 88 was placed in GenBank database under accession number SUB9024481 KO88 MW559226.

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
