# Peer review of "Understanding the Interactions between Staphylococcus aureus and the Raw-Meat-Processing Environment Isolate Klebsiella oxytoca in Dual-Species Biofilms via Discovering an Altered Metabolic Profile"

_microorganisms, 2021, doi:10.3390/microorganisms9040672_

Round 1
Reviewer 1 Report
Xiaozue Chen et al. “Understanding the interactions between Staphylococcus aureus and raw-meat-processing environment isolates Klebseilla oxytoca in dual-species biofilm via discovering altered metabolic profile”
This paper showed that biofilm formation increased when two species co-existed. In addition, the authors investigated metabolomics and transcriptomics. Although this study did not draw conclusive conclusion regarding dual-species biofilm formation, it shows quite meaningful results.
English should be revised.
Figure 1. Usually, growth over time uses a line graph. It is difficult to know the growth pattern of each strain over time with such a bar graph. We can identify the difference of growth of each strain every time with a line graph. I suggest to use line graph.
In addition, I don’t know the meaning of the letter ‘a, b, c, and d’ above the bar.
In this paper, Discussion part is lacking. Why?
Reviewer 2 Report
The authors have reported a significant interaction between S. aureus isolates and a coliform typically isolated from food processing facilities namely Klebsiella oxytoca. The authors describe the amplification of biofilm cell density as well as complex structure formation via microscopy and assays. Importantly, the authors describe alteration of biofilm metabolites during dual species biofilm formation by metabolomics. This is a well written paper.
The following are the major comments regarding the paper.
- While the authors have done a great job of describing their results they have not discussed their findings in terms of overall significance and impact. There are very interesting findings here with the enriched pathways as well as the cumulative nature of biofilm growth when these bacteria are co-cultured. However, its impact on food processing or how the two bacteria might be interacting to support each other nor the impact of the enriched pathways on the individual bacteria have been discussed here.
- While extensive experiments might be beyond the purview of the present submission it is important to help the reader understand how these findings are important. Suggest inclusion of a discussion section to address these issues.
- Also suggest investigating the growth / biofilm formation of S. aureus in bacterial conditioned media of K. oxytoca and vice versa to further look into whether the biofilm metabolites from one might support the other. This will also help to determine communication between the bacteria.
- I assume that the labels a, b, c above the bar graphs represent significant differences in Fig. 1. Suggest addition of the key in the figure legend.
- Please edit the figure titles to describe a phenomenon that is observed rather than the technique used.
- Suggest combining the positive and negative ion mode data to demonstrate the significantly different metabolites. This will add to the clarity since many metabolites are replicated in both modes (Fig. 3,4 and 5 and the tables).
Reviewer 3 Report
In the presented study authors investigated the dual-species interactions between Klebsiella oxytoca and Staphylococcus aureus – food-borne pathogens. The Authors used a well-designed methodology and the experiment findings are very interesting even though lack of discussion of the results. Generally, the manuscript is well written, but need some improvements Specific comments can be found below:
- Line 76 – how the Authors estimated the concentration of bacteria suspension?
- Figures 3,4 and 6 are unreadable, the text is too small
- There is no discussion of the results. Please add the “Discussion” section to your manuscript and compare your findings with the other Scientist’s work
Round 2
Reviewer 3 Report
The manuscript has been significantly improved. All comments have been taken into account by the authors and now the manuscript is acceptable.